

# Acute glyphosate exposure does not condition the response of microbial communities to a dry–rewetting disturbance in a soil with long history of glyphosate–based herbicides

Marco Allegrini[1], Elena del Valle Gomez[1], María Celina Zabaloy[2]

[1]Laboratorio de Biodiversidad Vegetal y Microbiana, IICAR–CONICET–UNR, Zavalla, 2125, Argentina
[2]Centro de Recursos Naturales Renovables de la Zona Semiárida (CERZOS–CONICET), Departamento de Agronomía, Universidad Nacional del Sur, Bahía Blanca, 8000, Argentina

*Correspondence to*: Marco Allegrini (marcalleg88@hotmail.com)

**Abstract.** Dry–rewetting perturbations are natural disturbances in the edaphic environment and particularly in dryland
cultivation areas. The interaction of this disturbance with glyphosate–based herbicides (GBHs) deserves special attention in the soil environment due to the intensification of agricultural practices and the acceleration of climate change with an intensified water cycle. The objective of this study was to assess the response of microbial communities in a soil with long history of GBHs to a secondary imposed perturbation (a single dry–rewetting event). A factorial microcosm study was conducted to evaluate the potential conditioning effect of an acute glyphosate exposure on the response to a following dry–
rewetting event. A Respiratory Quotient (RQ) based on an ecologically relevant substrate (*p*–coumaric acid) and basal respiration was used as physiological indicator. Similarly, DNA–based analyses were considered, including quantitative PCR (qPCR) of functional sensitive microbial groups linked to cycles of carbon (*Actinobacteria*) and nitrogen (ammonia–oxidizing microorganisms), qPCR of total bacteria and denaturing gradient gel electrophoresis (DGGE) of ammonia–oxidizing bacteria (AOB). Significant effects of Herbicide and of Dry–rewetting perturbations were observed in the RQ and
in the copy number of *amoA* gene of AOB, respectively. However, no significant interaction was observed between them when analyzing the physiological indicator and the copy number of the evaluated genes. PCR–DGGE results were not conclusive regarding a potential effect of Dry–rewetting × Herbicide interaction on AOB community structure, suggesting further analysis by deep sequencing of *amoA* gene.

## 1 Introduction

Soil microbial communities play a central role in several processes that contribute to a wide–range of important ecosystem services (Tilman et al., 2002; EFSA, 2016). Different factors with potential disruption effects on microbial communities and processes (e.g. pesticides), can reduce the functional sustainability of soils (Tilman et al., 2002). Among them, anthropic disturbances (e.g., pesticides) or natural disturbances like dry–rewetting events are common perturbations of the soil



environment, particularly in the context of global climate models which predict an intensification of the hydrological cycles

with more extended periods of droughts and more intense rainfalls (Huntington, 2006).

The effects of dry–rewetting cycles in the edaphic environment and on microbial communities have been considered in several studies (Hastings et al., 2000; Gleeson et al., 2008; Bustamante et al., 2012) Desiccation can affect microbial communities through nutritional limitation, osmotic stress and competition for available nutrients (Griffiths et al., 2003). Similarly, a rapid rewetting can trigger an osmotic shock inducing lysis, release of intracellular solutes and an increase in C

and N mineralization (Fierer et al., 2003). However, the interaction of these disturbances with the perturbation imposed by glyphosate–based herbicides (GBHs) has not been assessed before, even when the simultaneous exposure to both factors represents a common scenario in dryland cultivation areas such as in the semiarid Pampa of Argentina. These disturbance events could increase their frequency due to the intensification of agricultural practices based on glyphosate–resistant (GR) crops (Cerdeira and Duke, 2006) and repeated dry–rewetting cycles under an accelerating climate change (Huntington, 2006;

Evans and Wallenstein, 2011).

In a previous study, we reported no detection of a pollution–induced increase in microbial community tolerance (PICT) to glyphosate in a soil with long history of GBHs (Allegrini et al., 2015). Considering the aforementioned, we conducted a follow–up study to assess the response of microbial communities of a soil chronically exposed to GBHs to a secondary imposed perturbation (a single dry–rewetting event). The response of microbial communities to the perturbations imposed by

glyphosate exposure and dry–rewetting was assessed through a physiological indicator, calculated as the ratio of basal respiration to substrate induced respiration (SIR) with $p$–coumaric acid as amended substrate. This respiratory quotient (RQ) has demonstrated to be sensitive to repeated glyphosate applications (Allegrini et al., 2017). Similarly, DNA–based analyses were conducted to quantitate the abundance of genes from different microbial groups which could be affected by the imposed perturbations. We focused on microorganisms with well–known sensitivity to GBHs and other pesticides like

ammonia–oxidizing bacteria and archaea (AOB, AOA) (Zhang et al., 2018) and *Actinobacteria* (Barriuso et al., 2010). Ammonia oxidizing prokaryotes and Actinomycetes are involved in ecologically relevant processes in soil (N–cycling and organic matter turnover, respectively) and have been classified as microorganisms with high degree of sensitivity with respect to losses of organisms or functions (Anderson, 2003). We hypothesize that, if no increase in community tolerance was observed after long exposure to GBHs in the field, an acute exposure would not significantly modify the structure and

physiology of the microbial community so as to condition the sensitivity to a subsequent dry–rewetting disturbance.

## 2 Material and methods

### 2.1 Soil sampling and microcosm set up

Sampling was conducted in the same agricultural plot (ZAV$_H$) with long history of exposure to GBHs that was described in a previous study (Allegrini et al., 2015). Fifteen subsamples were taken at a 0–10 cm depth, sieved (<5.6 mm) and pooled to

obtain a composite sample. Soil was stored at 4°C and used within 6 days for the microcosm study.



Twelve microcosms (equivalent to 40 g of oven dry soil) were prepared in 100 ml sterile screw–cap polypropylene flasks, loosely capped to reduce water evaporation whilst leaving enough space for free passage of air. All flasks (60 % WHC) were pre–incubated in the dark at 25 °C (Ingelab I.501PF Incubator) for 1 week. Then, microcosms were randomly assigned to the following treatments, in a 2×2 factorial design with 3 replicates per treatment: "Herbicide" (two levels: with GBH "CG" and control with distilled sterile water "SG") and "Dry–rewetting" (two levels: with desiccation "CD" and untreated control "SD"). First, microcosms received either the CG or SG treatments (day 0). The herbicide (Roundup Full II, Monsanto™, N–(phosphonomethyl)glycine potassium salt, 66.2 % w v$^{-1}$, additives not specified) was applied in a final volume of 0.2 ml (with distilled water) at a rate of 49 µg active ingredient g$^{-1}$ soil similarly to other studies with silt loam soils (Haney et al., 2000; Ratcliff et al., 2006). This dose mimics the concentration of glyphosate found in soil after a 1× application rate in the field (0.84 kg ha$^{-1}$) considering a 2 mm soil interaction penetration due to the high absorptivity and low leachability of glyphosate (Haney et al., 2000). Microcosms were initially incubated for 14 days under conditions described above for the pre–incubation step. The dry–rewetting disturbance was imposed at day 14 and microcosms were returned to incubation for 14 days more. Sampling of microcosms for analysis was done on day 28. The dry–rewetting disturbance consisted of air–drying from the top with fan–forced air at room temperature (20–25 °C) during 24 h, followed by rewetting with distilled water up to 60 % WHC.

## 2.2 Physiological analysis

Substrate–induced respiration with *p*–coumaric acid and basal respiration in soil suspensions were determined with BD Oxygen Biosensor™ System microplates according to the same protocol and data processing details described in a previous study (Allegrini et al., 2017).

## 2.3 DNA–based analysis

### 2.3.1 DNA extraction and quantitation

The commercial kit PowerSoil™ DNA Isolation kit (MoBio, Inc., Carlsbad, CA) was used for DNA extraction from soil samples according to manufacturer instructions. DNA was quantified using QuantiFluor dsDNA kit in a Quantus fluorometer (Promega Madison, WI).

### 2.3.2 Quantification of indicator genes

Quantification of 16 rRNA gene, *amoA* gene of AOB (*amoA*$_{AOB}$) and *amoA* of AOA (*amoA*$_{AOA}$) was conducted by quantitative Real Time PCR (qPCR) using the protocols described in Allegrini et al. (2015), Zabaloy et al. (2016) and Zabaloy et al. (2017), respectively. For *Actinobacteria* the pair of primers S–P–Acti–1154–a–S–19/S–P–Acti–1339–a–A–18 was used (Pfeiffer et al., 2013). The composition of the master mix in the latter case was as follows: 7.5 µl of PCR iTaq Universal SYBR Green Supermix (2×; Bio–Rad Laboratories); 0.3 µl of each primer (stocks 10 µM, Invitrogen), 1 µl of



DNA (1–10 ng μl$^{-1}$) and ultrapure water to 15 μl. The amplification program was as follows: pre–incubation (95 °C, 5 min, 1 cycle), amplification (95 °C 15 s, 59 °C 30 s, 72 °C 45 s, 35 cycles), followed by melting curve analysis (65–95 °C). Decimal dilutions of a plasmid harboring one copy of 16S rRNA gene of *Streptomyces albus* DSM 40313 were used as standards (serial 10$^{-1}$ dilutions to obtain between $4.97 \times 10^6$ and $4.97 \times 10^2$ copies). All amplifications were conducted in ABI

7500 Real Time System (Applied Biosystems, Foster City, CA).

The abundance values of these genes were used as surrogates of population sizes, although no attempt was made to convert copies into cell numbers to avoid introducing errors (e.g. errors related with an unknown number of operons per cell in mixed bacterial communities) (Zabaloy et al., 2017; Ouyang et al., 2016).

**2.3.2 Denaturing gradient gel electrophoresis of AOB**

The amplification of *amoA*$_{AOB}$ with amoA–1F/amoA–2R primers (Rotthauwe et al., 1997) and the DGGE analysis of PCR products were conducted according to previously reported protocols (Allegrini et al., 2017). Digital gel images were processed with Software Gel Compare II$^{TM}$ v4.6 (Applied Maths). After optimization of gel properties normalization was conducted using amplicons of *Nitrosomonas europaea* and uncultured bacteria 5–A51 (accession number KJ643949 in GenBank) as internal reference positions (GelCompar II$^{TM}$ v. 4.6, Software Manual).

**2.4 Statistical analysis**

Respiratory quotient (RQ) values were analyzed using a two–way ANOVA at a 5 % significance level using R Statistical Software v3.5.0 (R Development Core team). The copy numbers of genes (log$_{10}$ copies μg$^{-1}$ DNA) were analyzed in the same way. In all cases, normality and homoscedasticity were verified with Shapiro–Wilks and Levene test, respectively (α=0.05).

Denaturing gradient gel electrophoresis fingerprints were analyzed with the Software GelCompar II$^{TM}$ v4.6 (Applied Maths, Kortrijk, Belgium) through cluster analysis using Pearson correlation coefficient (*r*) and Unweighted Pair Group Method with Arithmetic Mean (UPGMA) algorithm. Cophenetic correlation coefficients were calculated in each branch and the root to determine the quality of the dendrogram. Clusters were defined at 80 % similarity level (cut–off) and the 100 % internal stability of them (group separation assessment) was verified in GelCompar II using the statistical method Jackknife

resampling with average similarities (GelCompar II™ v. 4.6, Software Manual).

**3 Results**

**3.1 Respiratory responses**

The mean RQ values for the different treatments are indicated in Fig. 1. According to two–way ANOVA (Table 1), no interaction was observed between factors (*P* > 0.05). Thus, main effects were considered. No statistical significance was





observed for the main effect of Dry–rewetting. Conversely, Herbicide showed a significant effect ($P < 0.05$) with a higher RQ value in CG microcosms relative to the untreated microcosms (SG).

### 3.2 DNA–based analysis

### 3.2.1 Quantification of indicators genes

The equations obtained after linear regression of qPCR standard curves and the respective efficiencies are indicated in Table

2. Mean copy numbers for each treatment and each gene are shown in Fig. 2 and Table 4. For all the indicators genes, the results of two–way ANOVA (Table 3) indicated no statistical significance of Herbicide main effect as well as no interaction, while a significant Dry–rewetting effect was detected only for AOB ($P < 0.05$). The abundance of $amoA_{AOB}$ (averaged for both levels of Herbicide factor) was 1.27 fold higher in microcosms with dry–rewetting dessication (CD) than in undisturbed (SD) microcosms (Table 4).

### 3.2.2 DGGE of ammonia–oxidizing bacteria

DGGE profiles showed few bands and high similarity values (Pearson coefficients) among replicates of the four treatments, with no separation in four treatment–clusters. Similarly, no obvious separation was observed between microcosms with (CD) and without (SD) dry–rewetting or between glyphosate–treated (CG) and untreated microcosms (SG). At 80 % similarity level (cut–off), a separation in two clusters was observed (Fig. 3, grey branches). In one of them, we observed two replicates

of CD/SG treatment. In the second cluster the three replicates of CD/CG treatment clustered together with microcosms in which no dry–rewetting was applied (SD).

### 4 Discussion

In this study we evaluated whether an acute *in vitro* glyphosate application on a soil with long history of application of GBHs modulates the response of the microbial communities to the following dry–rewetting disturbance.

We hypothesized that if no PICT was observed after long exposure in the field (Allegrini et al., 2015), a single glyphosate application to microcosms would have no effect in the structure of the microbial community, as the probability to change to an alternative state is more likely in response to a press disturbance (chronic exposure) than to a pulse disturbance (Shade et al., 2012). Thus, the sensitivity to a secondary perturbation will not be conditioned by the presence/absence of a previous acute glyphosate exposure (Clements and Rohr, 2009). This hypothesis was confirmed by our results: no interaction was

observed between Herbicide and Dry–rewetting in an acute exposure to both perturbations with a physiological indicator (Table 1) and with DNA–based methods (Table 2), supporting the absence of a PICT response. The non–significant interaction observed for *Actinobacteria* (Table 2) indicates that one of the main characteristics of this microbial group, the high tolerance to desiccation (Evans and Wallestein, 2011), is not conditioned by the previous exposure to a single application of a GBH, even when negative effects of GBHs on this phylum have been reported (Barriuso et al., 2010). For



*amoA*, the absence of interaction is also a relevant observation considering that AOB are particularly sensitive to pesticides and also to water availability (Franzluebbers et al., 1995; Hastings et al., 2000; Gleeson et al., 2010). Thus, our results suggest that the sensitivity expected to each perturbation alone does not necessarily results in a synergic effect when combined.

Ammonia–oxidizing archaea were more abundant than AOB for all treatments. Also, they clearly differentiated from AOB
as no significant dry–rewetting effect was observed (Table 2). This observation is consistent with the results of Gleeson et al. (2010), who reported that AOB are more responsive to water availability than AOA. The statistical significance of dry–rewetting main effect on the abundance of AOB indicates that the microbial community of the soil assessed in this study is particularly sensitive to the perturbation. Conversely, the abundance of AOB seems to be less sensitive to GBH exposure (no significance detected for this factor), supporting previous results with the same soil and the same herbicide formulation in
which no effects of repeated applications were detected on absolute abundance (up to three applications) (Allegrini et al., 2017). As indicated in Table 3 and 4, the dry–rewetting perturbation enhanced the abundance of $amoA_{AOB}$ relative to the untreated microcosms (SD). Most gram negative bacteria are affected by a rapid rewetting after desiccation events and a recover to the initial abundance values has been reported for AOB at 18 days after rewetting (Hastings et al., 2000). At functional level (nitrification rate), Fierer and Schimel (2002) found a significant increase in the activity of autotrophic
nitrifying communities after several dry–rewetting cycles, in agreement with the higher abundance that we observed for $amoA_{AOB}$ and with a correlation between *amoA* copy number and nitrification potential observed in different soils (Rudisill et al., 2016; Zabaloy et al., 2017).

The low number of bands observed in the DGGE profiles of $amoA_{AOB}$ amplicons suggests a low richness of AOB in the studied soil. This result is in agreement with a previous biogeographic study which reported a low diversity of *amoA*
sequences in soil AOB communities, with most of them in the *Nitrosospira* lineages (Fierer et al., 2009). More recently, a microcosm study with a loam sandy soil from Pampa region observed low diversity in AOB community with DGGE (Zabaloy et al., 2017). An obvious separation among DGGE profiles of microcosms with and without dry–rewetting was not observed, indicating no effects of this perturbation on the community structure of AOB. Thus, even that qPCR indicated an increase in the abundance of $amoA_{AOB}$ sequences, the profiling (fingerprinting) of the community structure did not show the
same sensitivity to the dry–rewetting disturbance (Fig. 3).

The separation observed at 80 % similarity level (Fig. 3) between two replicates of CD/SG treatment and the three replicates CD/CG could be indicating an interaction as no comparable separation was detected between SD/SG and SD/CG. However, more evidences are still necessary to determine whether or not there is a significant interaction effect on the structure of AOB. Amplicon sequencing of $amoA_{AOB}$ and beta diversity analysis could provide substantially more information in this
regard.

In conclusion, our study demonstrates that acute exposure to a GBH does not have a conditioning effect on the response of microbial communities to a secondary disturbance (dry–rewetting) in a soil with chronic exposure to GBHs. To obtain more evidences supporting our conclusion, future studies should assess the effects of several dry–rewetting cycles.



## Author contribution

MA, MCZ and EG designed the experiment. MA and MCZ are credited for methodology, investigation and manuscript review and editing. MA conducted formal analysis and wrote the original draft. Project administration, resources and funding acquisition were conducted by EG and MCZ.

## Data availability

Data is available from 4TL Database (http://doi.org/10.4121/uuid:a86ce94c-1b3d-447a-8652-b2e2d0a72187). DOI:
10.4121/uuid:a86ce94c-1b3d-447a-8652-b2e2d0a72187.

## Acknowledgements

We acknowledge Technician A.M. Zamponi (CONICET) and trainee student D.A. Tebbe (Carl von Ossietzky Universität Oldenburg, Germany) for assistance with real time PCR analyses.

## Competing interests

The authors declare that they have no conflict of interest.

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



**Table 1.** Two–way ANOVA of respiratory quotient (RQ) values. The *P*–values indicated for the main effects of Herbicide and of Dry–rewetting disturbances correspond to the model without interaction as no significance ($P > 0.05$) was observed for this term. *df*: degrees of freedom.

| ANOVA RQ $_{p-\text{coumaric acid}}$ | |
| --- | --- |
| Dry–rewetting (*df* = 1) | *P = 0.34 (F = 1.01)* |
| Herbicide (*df* = 1) | ***P = 0.03** (F = 6.61)* |
| Interaction (*df* = 1) | *P = 0.92 (F = 0.01)* |
| Error *df* | 8 |


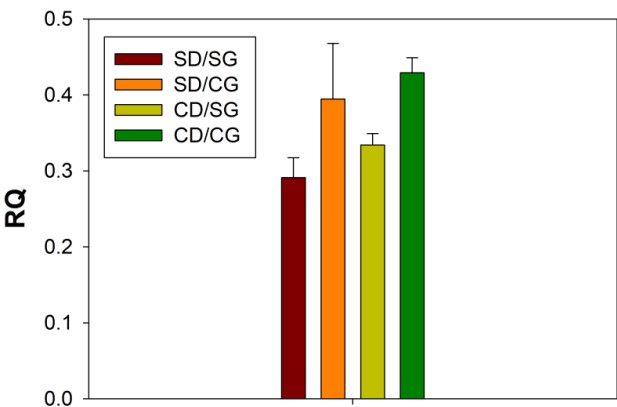

**Figure 1:** Respiratory quotient (RQ) values. The four treatments are indicated in different colours. Error bars indicate the standard error of the mean (*n*=3). SD/SG: No Dry–rewetting disturbance/No herbicide; SD/CG: No dry–rewetting disturbance/Herbicide; CD/SG: Dry–rewetting disturbance/No herbicide; CD/CG: Dry–rewetting disturbance/Herbicide.


**Table 2.** Equations of qPCR standard curves. The results for ammonia–oxidizing bacteria (AOB), ammonia–oxidizing archaea (AOA), *Actinobacteria* and total bacteria are indicated.

| Gene | Group | Equation | $R^2$ | Efficiency (%) |
| --- | --- | --- | --- | --- |
| *amoA* | AOB | $Ct = 41.21 - 3.76 \log_{10} \text{(copy number)}$ | 0.99 | 84.1 |
| *amoA* | AOA | $Ct = 38.19 - 3.56 \log_{10} \text{(copy number)}$ | 0.998 | 78.57 |
| 16S rRNA | Total bacteria | $Ct = 38.19 - 3.56 \log_{10} \text{(copy number)}$ | 0.999 | 91.07 |
| 16S rRNA | *Actinobacteria* | $Ct = 38.17 - 3.48 \log_{10} \text{(copy number)}$ | 1 | 93.67 |




**Table 3.** Two–way ANOVA of copy numbers for different indicator genes. The *P*–values indicated for the main effects of Herbicide and of Dry–rewetting disturbances correspond to the model without interaction as no significance ($P > 0.05$) was observed for this term. *df*: degrees of freedom.

| ANOVA | Total bacteria | *Actinobacteria* | AOB | AOA |
|---|---|---|---|---|
| Dry–rewetting (*df* = 1) | $P = 0.42$ | $P = 0.13$ | *$P = 0.026$ | $P = 0.06$ |
| Herbicide (*df* =1) | $P = 0.97$ | $P = 0.63$ | $P = 0.57$ | $P = 0.83$ |
| Interaction (*df*=1) | $P = 0.52$ | $P = 0.68$ | $P = 0.88$ | $P = 0.97$ |
| Error *df* | 8 | 8 | 8 | 8 |

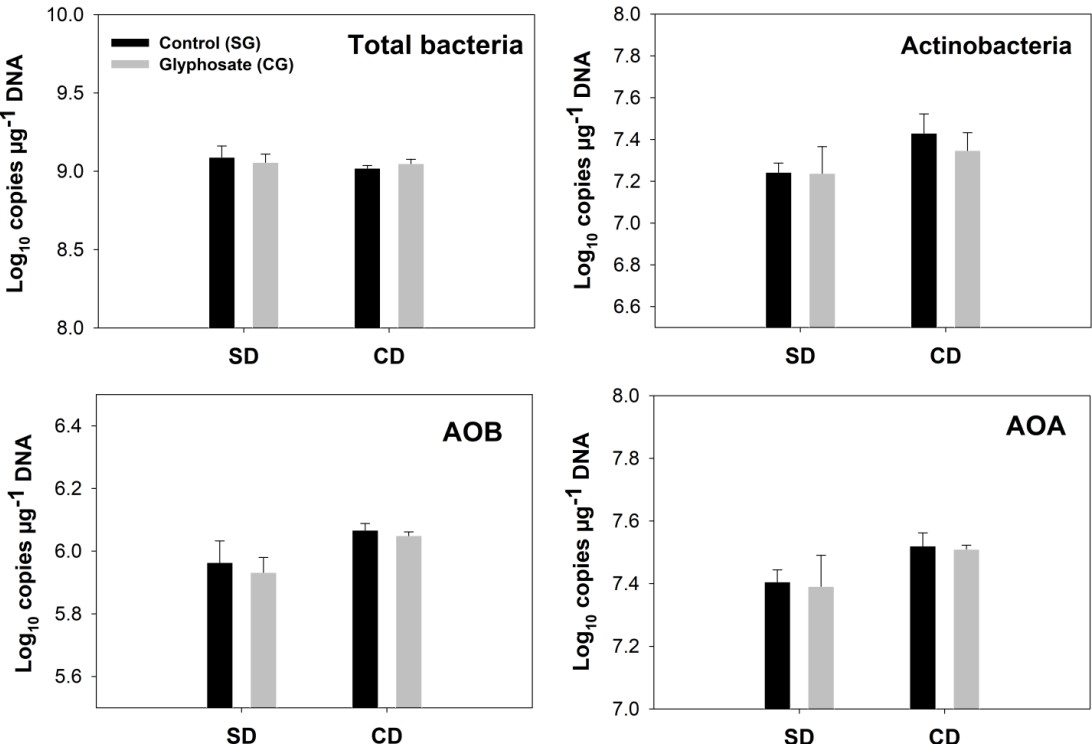

**Figure 2.** Copy number of indicator genes for total bacteria, *Actinobacteria*, AOB and AOA. Error bars indicate the standard error of the mean (*n*=3). SD/SG: No dry–rewetting disturbance/No herbicide; SD/CG: No dry–rewetting disturbance/Herbicide; CD/SG: Dry–rewetting disturbance/No herbicide; CD/CG: Dry–rewetting disturbance/Herbicide.



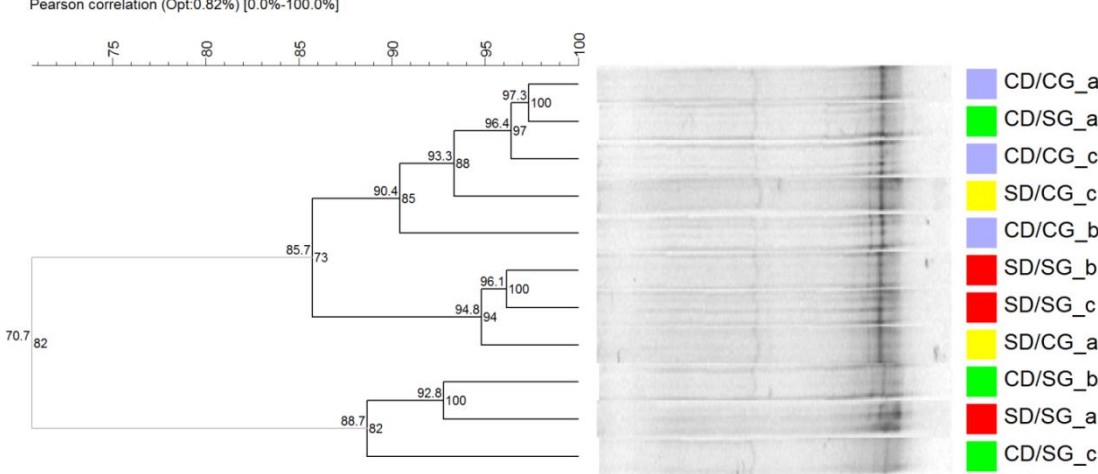

**Figure 3.** Cluster analysis of DGGE profiles of AOB. The dendrogram was obtained using Pearson–UPGMA analysis of densitometric profiles. Treatments are indicated in different colours. Lower case letters indicate replicates within treatments. In each node, the left number indicates the similarity value ($r \times 100$), while the right number is the cophenetic correlation coefficient. Grey branches indicate clusters with 100 % internal stability according to Jackknife method, defined at 80 % similarity value. SD/SG: No dry–rewetting disturbance/No herbicide; SD/CG: No dry–rewetting disturbance/Herbicide; CD/SG: Dry–rewetting disturbance/No herbicide; CD/CG: Dry–rewetting disturbance/Herbicide.

**Table 4. Copy number (copies µg$^{-1}$ DNA) of the indicator genes assessed for the different microbial groups.** SD/SG: No dry–rewetting/No herbicide; SD/CG: no dry–rewetting disturbance/Herbicide; CD/SG: dry–rewetting disturbance/no herbicide; CD/SG: Dry–rewetting disturbance/Herbicide.

| Treatment | AOB | AOA | Total bacteria | *Actinobacteria* | AOB* |
|---|---|---|---|---|---|
| SD/SG | $9.44 \times 10^5 \pm 1.60 \times 10^5$ | $2.56 \times 10^7 \pm 2.24 \times 10^6$ | $1.26 \times 10^9 \pm 1.99 \times 10^8$ | $1.77 \times 10^7 \pm 1.86 \times 10^6$ | $9.05 \times 10^5 \pm 8.47 \times 10^4$ (SD) |
| SD/CG | $8.66 \times 10^5 \pm 9.32 \times 10^4$ | $2.59 \times 10^7 \pm 5.50 \times 10^6$ | $1.16 \times 10^9 \pm 1.47 \times 10^8$ | $1.90 \times 10^7 \pm 6.01 \times 10^6$ | |
| CD/SG | $1.17 \times 10^6 \pm 5.84 \times 10^4$ | $3.34 \times 10^7 \pm 3.17 \times 10^6$ | $1.05 \times 10^9 \pm 4.05 \times 10^7$ | $2.81 \times 10^7 \pm 5.22 \times 10^6$ | $1.15 \times 10^6 \pm 3.16 \times 10^4$ (CD) |
| CD/CG | $1.12 \times 10^6 \pm 3.15 \times 10^4$ | $3.24 \times 10^7 \pm 9.59 \times 10^5$ | $1.13 \times 10^9 \pm 6.81 \times 10^7$ | $2.31 \times 10^7 \pm 7.49 \times 10^6$ | |

*Copy number of microcosms with (CD) or without (SD) dry–rewetting disturbance averaged through all levels of Herbicide factor.