# Peer review of "Acute glyphosate exposure does not condition the response of microbial communities to a dry–rewetting disturbance in a soil with long history of glyphosate–based herbicides"

_SOIL, 2020_

## Referee Comment (RC1) · Anonymous Referee #1 · 22 Apr 2020

General comments: In this manuscript, the authors evaluate the potential conditioning effect of an acute glyphosate exposure (first imposed perturbation) on the response of soil microbial communities to a single dry-rewetting event (second imposed pertur-bation), in soils with a long history of exposure to glyphosate-based herbicides. The topic under study is relevant, the hypothesis is sound, and the experimental design is suitable for the aim of the study. In addition, the manuscript is concise, well written and organized; therefore I recommend its publication after some minor revisions.

[Figure]

Technical corrections:

L27-28: The phrase "(e.g. pesticides)" is repeated in both sentences; maybe it's not necessary to mention it twice.

L124-125 and Table 2: I'm not sure that this table is really necessary here. Maybe the information of R2 and %Efficiency could be simply put in a sentence in the methods section? I suggest a brief sentence, like: "The efficiencies of qPCR assays were 84.1% (amoA-AOB), 78.57% (amoA-AOA), 91.07% (total bacteria 16S rRNA) and 93.67% (Actinobacteria 16S rRNA); and R2 values were $\geq$ 0.99 in all assays".

L145-146: I'm not sure that I'm getting this right. How does the lack of interaction between the two disturbances support the absence of a PICT response? Can you briefly clarify what a clear PICT response would be? Is it possible that even if there was a PICT response, there wasn't interaction with the second perturbation (desiccation)?

L147 and L155: Can you please check Table numbers? I believe it's Table 3.

L152: "does not necessarily result"

L161: Tables 3 and 4.

L173: Maybe "even though" instead of "even that"?

Figure 3: I'm sorry, what do lowercase letters mean? Replicates within treatments sometimes have different lowercase letters, e.g., CD/SG_a, CD/SG_b and CD/SG_c.

---

## Referee Comment (RC2) · Anonymous Referee #2 · 15 May 2020

The response of microbial communities to different perturbations is of great interest for designing sustainable farming practices (either tactic or strategic). Particularly the long term effect of GBHs is relevant in no-till agricultural systems, and the dry-wetting effects are important in rainfed agriculture. This research explores in a microcosm experiment the effect of GBHs and dry-rewetting perturbations on soil microbial communities, but the interaction effect was not clear. Despite sound methods were used in the present work, deeper studies are needed and can be addressed with new research techniques

like microbiome sequencing and also by repeated cycles of dry-rewetting to address more clearly the ecological impact (eg. resilience, resistance to disturbance). The manuscript is appropriate for publishing in SOIL. Some minor corrections are needed: 1- Check references: year in text is different from the year in References list a. Line 40 and 148: Evans and Wallenstein, 2011 or 2012? b. Line 87: Zabaloy et al 2016 is not in Reference list c. Line 89: Pfeiffer et al 2013 or 2014? d. Line 144: Clements and Rohr 2009 or 2008? e. Line 151: Franzluebbers et al 1995 or 1994?

2- As Reviewer #1 suggests, the concept of PICT response and the absence of inter-action could be explained with more detail.

3- Line 58: how many years is "long term"? Despite described in Allegrini et al 2015, please indicate in the text.

4- Line 48: change quantitae by quantity

---

## Author Comment (AC1) · 21 May 2020

Anonymous Referee #1 General comments: In this manuscript, the authors evaluate the potential conditioning effect of an acute glyphosate exposure (first imposed perturbation) on the response of soil microbial communities to a single dry-rewetting event (second imposed perturbation), in soils with a long history of exposure to glyphosate-based herbicides. The topic under study is relevant, the hypothesis is sound, and the experimental design is suitable for the aim of the study. In addition, the manuscript is concise, well written and organized; therefore I recommend its publication after some minor revisions.

Technical corrections: L27-28: The phrase "(e.g. pesticides)" is repeated in both sentences; maybe it's not necessary to mention it twice. AC: We agree with the comment. The phrase will be removed in line 27.

L124-125 and Table 2: I'm not sure that this table is really necessary here. Maybe the information of R2 and %Efficiency could be simply put in a sentence in the methods section? I suggest a brief sentence, like: "The efficiencies of qPCR assays were 84.1% (amoA-AOB), 78.57% (amoA-AOA), 91.07% (total bacteria 16S rRNA) and 93.67% (Actinobacteria 16S rRNA); and R2 values were ≥ 0.99 in all assays". AC: We agree with the suggestion. Table 2 will be removed and the information will be inserted in the text in the same way as suggested by the reviewer.

L145-146: I'm not sure that I'm getting this right. How does the lack of interaction between the two disturbances support the absence of a PICT response? Can you briefly clarify what a clear PICT response would be? Is it possible that even if there was a PICT response, there wasn't interaction with the second perturbation (desiccation)? AC: Changes in microbial communities associated with the development of a greater tolerance (PICT) to a pesticide in the field (chronic exposure) might, at the same time, conceal a higher sensitivity in the response to other perturbations (a "cost of tolerance" when adapting to an environmental stress; Clements and Rohr, 2009). Thus, if no PICT response was observed in the studied soil after long exposure in the field (Allegrini et al., 2015), it could be expected that a single glyphosate application to microcosms (acute exposure) would have no effect at all in the structure of the microbial community and, consequently, no conditioning effect of this acute glyphosate exposure should be observed on the response to a secondary perturbation (dry-rewetting in this case). The absence of conditioning effect is consistently reflected in the non-significant interaction term of ANOVA. However, it is important to mention that even if a PICT response would

have been observed in this soil, the higher tolerance could have associated costs in the response to only some environmental stresses (e.g., to stresses caused by other xenobiotics but no to a dry-rewetting stress). Thus, a non-significant interaction could be also observed for a soil in which a PICT has been detected. Based on this argument, we consider that the absence of interaction in our study is not a conclusive result supporting the absence of a PICT response. In other words, the result we observed in the microcosm assay (no conditioning effect of an acute glyphosate exposure to dry-rewetting response) is an expected result for a soil in which no PICT was observed (as explained above) but it cannot be considered a supporting evidence of the absence of a PICT response in this soil. We will remove the phrase "supporting the absence of a PICT response" in line 146. Also we will introduce the concept of "cost of tolerance" after line 142, as explained before in response to the reviewer comment.

L147 and L155: Can you please check Table numbers? I believe it's Table 3. AC: As indicated by the reviewer it is Table 3 and not Table 2.

L152: "does not necessarily result" AC: Ok, the error will be corrected.

L161: Tables 3 and 4. AC: Ok, the error will be corrected.

L173: Maybe "even though" instead of "even that"? AC: We agree with the suggestion. Figure 3: I'm sorry, what do lowercase letters mean? Replicates within treatments sometimes have different lowercase letters, e.g., CD/SG_a, CD/SG_b and CD/SG_c. AC: Lowercase letters were used to identify the different replicates within treatments.

Anonymous Referee #2 The response of microbial communities to different perturbations is of great interest for designing sustainable farming practices (either tactic or strategic). Particularly the long term effect of GBHs is relevant in no-till agricultural systems, and the dry-wetting effects are important in rain-fed agriculture. This research explores in a microcosm experiment the effect of GBHs and dry-rewetting perturbations on soil microbial communities, but the interaction effect was not clear. Despite sound methods were used in the present work, deeper studies are needed and can be addressed with new research techniques like microbiome sequencing and also by repeated cycles of dry-rewetting to address more clearly the ecological impact (eg. resilience, resistance to disturbance). The manuscript is appropriate for publishing in SOIL. Some minor corrections are needed:

1- Check references: year in text is different from the year in References list a. Line 40 and 148: Evans and Wallenstein, 2011 or 2012? b. Line 87: Zabaloy et al 2016 is not in Reference list c. Line 89: Pfeiffer et al 2013 or 2014? d. Line 144: Clements and Rohr 2009 or 2008? e. Line 151: Franzluebbers et al 1995 or 1994? AC: All references were checked and the modifications will be introduced as indicated by the reviewer.

2- As Reviewer #1 suggests, the concept of PICT response and the absence of interaction could be explained with more detail. AC: We agree with the need of clarification of this concept. Please see the response to the third comment of Reviewer 1 (L145-146).

3- Line 58: how many years is "long term"? Despite described in Allegrini et al 2015, please indicate in the text. AC: With long-term we refer to a history of more than 20 years. We will introduce it in the text as suggested by the reviewer.

4- Line 48: change quantitae by quantity. AC: The change will be introduced in the text as indicated by the reviewer.

---

## Author Response (AR1)

**Acute glyphosate exposure does not condition the response of microbial communities to a dry–rewetting disturbance in a soil with long history of glyphosate–based herbicides**

Marco Allegrini[1], Elena del Valle Gomez[1], María Celina Zabaloy[2]

[1]Laboratorio de Biodiversidad Vegetal y Microbiana, IICAR–CONICET–UNR, Zavalla, 2125, Argentina
[2]Centro de Recursos Naturales Renovables de la Zona Semiárida (CERZOS–CONICET), Departamento de Agronomía, Universidad Nacional del Sur, Bahía Blanca, 8000, Argentina

*Correspondence to*: Marco Allegrini (marcalleg88@hotmail.com)

**Abstract.** Dry–rewetting perturbations are natural disturbances in the edaphic environment and particularly in dryland cultivation areas. The interaction of this disturbance with glyphosate–based herbicides (GBHs) deserves special attention in the soil environment due to the intensification of agricultural practices and the acceleration of climate change with an intensified water cycle. The objective of this study was to assess the response of microbial communities in a soil with long history of GBHs to a secondary imposed perturbation (a single dry–rewetting event). A factorial microcosm study was conducted to evaluate the potential conditioning effect of an acute glyphosate exposure on the response to a following dry–rewetting event. A Respiratory Quotient (RQ) based on an ecologically relevant substrate (*p*–coumaric acid) and basal respiration was used as physiological indicator. Similarly, DNA–based analyses were considered, including quantitative PCR (qPCR) of functional sensitive microbial groups linked to cycles of carbon (*Actinobacteria*) and nitrogen (ammonia–oxidizing microorganisms), qPCR of total bacteria and denaturing gradient gel electrophoresis (DGGE) of ammonia–oxidizing bacteria (AOB). Significant effects of Herbicide and of Dry–rewetting perturbations were observed in the RQ and in the copy number of *amoA* gene of AOB, respectively. However, no significant interaction was observed between them when analyzing the physiological indicator and the copy number of the evaluated genes. PCR–DGGE results were not conclusive regarding a potential effect of Dry–rewetting × Herbicide interaction on AOB community structure, suggesting further analysis by deep sequencing of *amoA* gene. The results of this study indicate that the perturbation of an acute glyphosate exposure in a soil with long–history of this herbicide does not have a conditioning effect on the response to a subsequent dry–rewetting disturbance according to a physiological indicator or the quantified bacterial/archaeal genes.

[revised manuscript text omitted]

**2.3.2 Denaturing gradient gel electrophoresis of AOB**

The amplification of $amoA_{AOB}$ with amoA–1F/amoA–2R primers (Rotthauwe et al., 1997) and the DGGE analysis of PCR

105    products were conducted according to previously reported protocols (Allegrini et al., 2017). Digital gel images were processed with Software Gel Compare II$^{TM}$ v4.6 (Applied Maths). After optimization of gel properties normalization was conducted using amplicons of *Nitrosomonas europaea* and uncultured bacteria 5–A51 (accession number KJ643949 in GenBank) as internal reference positions (GelCompar II$^{TM}$ v. 4.6, Software Manual).

**2.4 Statistical analysis**

110    Respiratory quotient (RQ) values were analyzed using a two–way ANOVA at a 5 % significance level using R Statistical Software v3.5.0 (R Development Core team). The copy numbers of genes ($\log_{10}$ copies μg$^{-1}$ DNA) were analyzed in the same way. In all cases, normality and homoscedasticity were verified with Shapiro–Wilks and Levene test, respectively (α=0.05).

Denaturing gradient gel electrophoresis fingerprints were analyzed with the Software GelCompar II$^{TM}$ v4.6 (Applied Maths,

115    Kortrijk, Belgium) through cluster analysis using Pearson correlation coefficient (*r*) and Unweighted Pair Group Method with Arithmetic Mean (UPGMA) algorithm. Cophenetic correlation coefficients were calculated in each branch and the root to determine the quality of the dendrogram. Clusters were defined at 80 % similarity level (cut–off) and the 100 % internal

stability of them (group separation assessment) was verified in GelCompar II using the statistical method Jackknife resampling with average similarities (GelCompar II™ v. 4.6, Software Manual).

**3 Results**

**3.1 Respiratory responses**

The mean RQ values for the different treatments are indicated in Fig. 1. According to two–way ANOVA (Table 1), no interaction was observed between factors ($P > 0.05$). Thus, main effects were considered. No statistical significance was observed for the main effect of Dry–rewetting. Conversely, Herbicide showed a significant effect ($P < 0.05$) with a higher RQ value in CG microcosms relative to the untreated microcosms (SG).

**3.2 DNA–based analysis**

**3.2.1 Quantification of indicators genes**

For all the indicators genes, the results of two–way ANOVA (Table 32) indicated no statistical significance of Herbicide main effect as well as no interaction, while a significant Dry–rewetting effect was detected only for AOB ($P < 0.05$). The equations obtained after linear regression of qPCR standard curves and the respective efficiencies are indicated in Table 2. Mean copy numbers for each treatment and each gene are shown in Fig. 2 and Table 43. For all the indicators genes, the results of two–way ANOVA (Table 3) indicated no statistical significance of Herbicide main effect as well as no interaction, while a significant Dry–rewetting effect was detected only for AOB ($P < 0.05$). The abundance of $amoA_{AOB}$ (averaged for both levels of Herbicide factor) was 1.27 fold higher in microcosms with dry–rewetting dessication (CD) than in undisturbed (SD) microcosms (Table 43).

**3.2.2 DGGE of ammonia–oxidizing bacteria**

DGGE profiles showed few bands and high similarity values (Pearson coefficients) among replicates of the four treatments, with no separation in four treatment–clusters. Similarly, no obvious separation was observed between microcosms with (CD) and without (SD) dry–rewetting or between glyphosate–treated (CG) and untreated microcosms (SG). At 80 % similarity level (cut–off), a separation in two clusters was observed (Fig. 3, grey branches). In one of them, we observed two replicates of CD/SG treatment. In the second cluster the three replicates of CD/CG treatment clustered together with microcosms in which no dry–rewetting was applied (SD).

**4 Discussion**

In this study we evaluated whether an acute *in vitro* glyphosate application on a soil with long history of application of GBHs modulates the response of the microbial communities to the following dry–rewetting disturbance.

We hypothesized that if no PICT was observed in the studied soil after long exposure in the field (Allegrini et al., 2015), a single glyphosate application to microcosms would have no effect in the structure of the microbial community, as the probability to change to an alternative state is more likely in response to a press disturbance (chronic exposure) than to a pulse disturbance (Shade et al., 2012). These changes in microbial communities associated with greater tolerance to a pesticide might, at the same time, conceal a higher sensitivity in the response to other perturbations (a "cost of tolerance"; Clements and Rohr, 2009). Thus, for the soil assessed in this study, we expected no conditioning effect in the  sensitivity to a secondary perturbation  
[revised manuscript text omitted]

350

355

360

365

370

375  **General comments: In this manuscript, the authors evaluate the potential conditioning effect of an acute glyphosate exposure (first imposed perturbation) on the response of soil microbial communities to a single dry-rewetting event (second imposed perturbation), in soils with a long history of exposure to glyphosate-based herbicides. The topic under study is relevant, the hypothesis is sound, and the experimental design is suitable for the aim of the study. In addition, the manuscript is concise, well written and organized; therefore I recommend its publication after some**

380  **minor revisions.**

**Technical corrections:**

**L27-28: The phrase "(e.g. pesticides)" is repeated in both sentences; maybe it's not necessary to mention it twice.**

We agree with the comment. The phrase was removed in line 27.

**L124-125 and Table 2: I'm not sure that this table is really necessary here. Maybe the information of R2 and**

385  **%Efficiency could be simply put in a sentence in the methods section? I suggest a brief sentence, like: "The efficiencies of qPCR assays were 84.1% (amoA-AOB), 78.57% (amoA-AOA), 91.07% (total bacteria 16S rRNA) and 93.67% (Actinobacteria 16S rRNA); and R2 values were ≥ 0.99 in all assays".**

We agree with the suggestion. Table 2 was removed and the information was inserted in the text in the same way as suggested by the reviewer.

390  **L145-146: I'm not sure that I'm getting this right. How does the lack of interaction between the two disturbances support the absence of a PICT response? Can you briefly clarify what a clear PICT response would be? Is it possible that even if there was a PICT response, there wasn't interaction with the second perturbation (desiccation)?**

Changes in microbial communities associated with the development of a greater tolerance (PICT) to a pesticide in the field (chronic exposure) might, at the same time, conceal a higher sensitivity in the response to other perturbations (a "cost of

395  tolerance" when adapting to an environmental stress; Clements and Rohr, 2009). Thus, if no PICT response was observed in the studied soil after long exposure in the field (Allegrini et al., 2015), it could be expected that a single glyphosate application to microcosms (acute exposure) would have no effect at all in the structure of the microbial community and, consequently, no conditioning effect of this acute glyphosate exposure should be observed on the response to a secondary perturbation (dry-rewetting in this case). The absence of conditioning effect is consistently reflected in the non-significant

400  interaction term of ANOVA.

However, it is important to mention that even if a PICT response would have been observed in this soil, the higher tolerance could have associated costs in the response to only some environmental stresses (e.g., to stresses caused by other xenobiotics but no to a dry-rewetting stress). Thus, a non-significant interaction could be also observed for a soil in which a PICT has been detected. Based on this argument, we consider that the absence of interaction in our study is not a conclusive result

405  supporting the absence of a PICT response. **In other words, the result we observed in the microcosm assay (no**

**conditioning effect of an acute glyphosate exposure to dry-rewetting response) is an expected result for a soil in which no PICT was observed (as explained above) but it cannot be considered a supporting evidence of the absence of a PICT response in this soil.**

We have removed the phrase "supporting the absence of a PICT response" in line 146. Also we have introduced the concept of "cost of tolerance" after line 142, as explained before in response to the reviewer comment.

**L147 and L155: Can you please check Table numbers? I believe it's Table 3.**

As indicated by the reviewer it is Table 3 and not Table 2.

**L152: "does not necessarily result"**

Ok, the error was corrected.

**L161: Tables 3 and 4.**

Ok, the error was corrected.

**L173: Maybe "even though" instead of "even that"?**

We agree with the suggestion.

**Figure 3: I'm sorry, what do lowercase letters mean? Replicates within treatments sometimes have different lowercase letters, e.g., CD/SG_a, CD/SG_b and CD/SG_c.**

Lowercase letters were used to identify the different replicates within treatments.

**Anonymous Referee #2**

**The response of microbial communities to different perturbations is of great interest for designing sustainable farming practices (either tactic or strategic). Particularly the long term effect of GBHs is relevant in no-till agricultural systems, and the dry-wetting effects are important in rainfed agriculture. This research explores in a microcosm experiment the effect of GBHs and dry-rewetting perturbations on soil microbial communities, but the interaction effect was not clear. Despite sound methods were used in the present work, deeper studies are needed and can be addressed with new research techniques like microbiome sequencing and also by repeated cycles of dry-rewetting to address more clearly the ecological impact (eg. resilience, resistance to disturbance). The manuscript is appropriate for publishing in SOIL. Some minor corrections are needed:**

**1- Check references: year in text is different from the year in References list a. Line 40 and 148: Evans and Wallenstein, 2011 or 2012? b. Line 87: Zabaloy et al 2016 is not in Reference list c. Line 89: Pfeiffer et al 2013 or 2014? d. Line 144: Clements and Rohr 2009 or 2008? e. Line 151: Franzluebbers et al 1995 or 1994?**

All references were checked and the modifications were introduced as indicated by the reviewer.

**2- As Reviewer #1 suggests, the concept of PICT response and the absence of interaction could be explained with more detail.**

We agree with the need of clarification of this concept. Please see the response to the third comment of Reviewer 1 (L145-146).

**3- Line 58: how many years is "long term"? Despite described in Allegrini et al 2015, please indicate in the text.**

440 With long-term we refer to a history of more than 20 years. We introduced it in the text as suggested by the reviewer.

**4- Line 48: change quantitae by quantity.**

The change was introduced in the text as indicated by the reviewer.

**List of all relevant changes**

All relevant changes were introduced in response to the reviewers' comments and were indicated previously.

445 In addition to these changes, the following changes were also introduced:

**Abstract (Lines 23 to 25 of the revised version of the manuscript)**

The following lines were introduced in response to the Topical Editor comments:

"The results of this study indicate that the perturbation of an acute glyphosate exposure in a soil with long-history of this herbicide does not have a conditioning effect on the response to a subsequent dry-rewetting disturbance according to a

450 physiological indicator or the quantified bacterial/archaeal genes."

**Data availability**

A typing error was found in the name of the database and was corrected:

"4TL Database" changed by "4TU Research.Data Database"

---

## Author Response (AR2)

June 11, 2020

Executive Editor
Soil
Dr. Jeanette Whitaker,

We are pleased to submit the requested revisions of the manuscript entitled "Acute glyphosate exposure does not condition the response of microbial communities to a dry–rewetting disturbance in a soil with long history of glyphosate–based herbicides". We are deeply grateful to both reviewers and yourself for giving us the opportunity to
considerably improve our manuscript with many useful comments.

We look forward to receiving yours feedback,

Sincerely Yours,

Dr. Marco Allegrini (Corresponding author)

E-mail address: marcalleg88@hotmail.com

**Acute glyphosate exposure does not condition the response of microbial communities to a dry–rewetting disturbance in a soil with long history of glyphosate–based herbicides**

Marco Allegrini[1], Elena del Valle Gomez[1], María Celina Zabaloy[2]

[1]Laboratorio de Biodiversidad Vegetal y Microbiana, IICAR–CONICET–UNR, Zavalla, 2125, Argentina
[2]Centro de Recursos Naturales Renovables de la Zona Semiárida (CERZOS–CONICET), Departamento de Agronomía, Universidad Nacional del Sur, Bahía Blanca, 8000, Argentina

*Correspondence to*: Marco Allegrini (marcalleg88@hotmail.com)

**Abstract.** Dry–rewetting perturbations are natural disturbances in the edaphic environment and particularly in dryland cultivation areas. The interaction of this disturbance with glyphosate–based herbicides (GBHs) deserves special attention in the soil environment due to the intensification of agricultural practices and the acceleration of climate change with an intensified water cycle. The objective of this study was to assess the response of microbial communities in a soil with long history of GBHs to a secondary imposed perturbation (a single dry–rewetting event). A factorial microcosm study was conducted to evaluate the potential conditioning effect of an acute glyphosate exposure on the response to a following dry–rewetting event. A Respiratory Quotient (RQ) based on an ecologically relevant substrate (*p*–coumaric acid) and basal respiration was used as physiological indicator. Similarly, DNA–based analyses were considered, including quantitative PCR (qPCR) of functional sensitive microbial groups linked to cycles of carbon (*Actinobacteria*) and nitrogen (ammonia–oxidizing microorganisms), qPCR of total bacteria and denaturing gradient gel electrophoresis (DGGE) of ammonia–oxidizing bacteria (AOB). Significant effects of Herbicide and of Dry–rewetting perturbations were observed in the RQ and in the copy number of *amoA* gene of AOB, respectively. However, no significant interaction was observed between them when analyzing the physiological indicator and the copy number of the evaluated genes. PCR–DGGE results were not conclusive regarding a potential effect of Dry–rewetting × Herbicide interaction on AOB community structure, suggesting further analysis by deep sequencing of *amoA* gene. The results of this study indicate that the perturbation of an acute glyphosate exposure in a soil with long–history of this herbicide does not have a conditioning effect on the response to a subsequent dry–rewetting disturbance according to a physiological indicator or the quantified bacterial/archaeal genes. This is particularly relevant for the sustainability of soils in rainfed agriculture, where frequent exposure to GBHs along with intensification of hydrological cycles are expected to occur. Further studies considering multiple dry–rewetting disturbances and in different soil types should be conducted to simulate those conditions and to validate our results.

[revised manuscript text omitted]

In conclusion, our study demonstrates that acute exposure to a GBH does not have a conditioning effect on the response of microbial communities to a secondary disturbance (dry–rewetting) in a soil with chronic exposure to GBHs. In a global context of extended use of GBHs and climate models which predict an intensification of hydrological cycles, our results are particularly relevant for the sustainability of soils in rainfed agriculture, where dry-rewetting cycles and GBHs applications are expected to occur simultaneously. To obtain more evidences supporting our conclusion, future studies should assess the effects of several dry–rewetting cycles and in different soil types.

**Author contribution**

MA, MCZ and EG designed the experiment. MA and MCZ are credited for methodology, investigation and manuscript review and editing. MA conducted formal analysis and wrote the original draft. Project administration, resources and funding acquisition were conducted by EG and MCZ.

**Data availability**

Data is available from  4TU.ResearchData Database (http://doi.org/10.4121/uuid:a86ce94c-1b3d-447a-8652- b2e2d0a72187). DOI: 10.4121/uuid:a86ce94c-1b3d-447a-8652-b2e2d0a72187.

**Acknowledgements**

[revised manuscript text omitted]

 Point-by-point response to the reviewers' comments

Anonymous Referee #1

**General comments: In this manuscript, the authors evaluate the potential conditioning effect of an acute glyphosate exposure (first imposed perturbation) on the response of soil microbial communities to a single dry-rewetting event (second imposed perturbation), in soils with a long history of exposure to glyphosate-based herbicides. The topic under study is relevant, the hypothesis is sound, and the experimental design is suitable for the aim of the study. In addition, the manuscript is concise, well written and organized; therefore I recommend its publication after some minor revisions.**

**Technical corrections:**

**L27-28: The phrase "(e.g. pesticides)" is repeated in both sentences; maybe it's not necessary to mention it twice.**

We agree with the comment. The phrase was removed in line 27.

**L124-125 and Table 2: I'm not sure that this table is really necessary here. Maybe the information of R2 and %Efficiency could be simply put in a sentence in the methods section? I suggest a brief sentence, like: "The efficiencies of qPCR assays were 84.1% (amoA-AOB), 78.57% (amoA-AOA), 91.07% (total bacteria 16S rRNA) and 93.67% (Actinobacteria 16S rRNA); and R2 values were ≥ 0.99 in all assays".**

We agree with the suggestion. Table 2 was removed and the information was inserted in the text in the same way as suggested by the reviewer.

**L145-146: I'm not sure that I'm getting this right. How does the lack of interaction between the two disturbances support the absence of a PICT response? Can you briefly clarify what a clear PICT response would be? Is it possible that even if there was a PICT response, there wasn't interaction with the second perturbation (desiccation)?**

Changes in microbial communities associated with the development of a greater tolerance (PICT) to a pesticide in the field (chronic exposure) might, at the same time, conceal a higher sensitivity in the response to other perturbations (a "cost of tolerance" when adapting to an environmental stress; Clements and Rohr, 2009). Thus, if no PICT response was observed in the studied soil after long exposure in the field (Allegrini et al., 2015), it could be expected that a single glyphosate application to microcosms (acute exposure) would have no effect at all in the structure of the microbial community and, consequently, no conditioning effect of this acute glyphosate exposure should be observed on the response to a secondary perturbation (dry-rewetting in this case). The absence of conditioning effect is consistently reflected in the non-significant interaction term of ANOVA.

However, it is important to mention that even if a PICT response would have been observed in this soil, the higher tolerance could have associated costs in the response to only some environmental stresses (e.g., to stresses caused by other xenobiotics but no to a dry-rewetting stress). Thus, a non-significant interaction could be also observed for a soil in which a PICT has been detected. Based on this argument, we consider that the absence of interaction in our study is not a conclusive result supporting the absence of a PICT response. **In other words, the result we observed in the microcosm assay (no conditioning effect of an acute glyphosate exposure to dry-rewetting response) is an expected result for a soil in which no PICT was observed (as explained above) but it cannot be considered a supporting evidence of the absence of a**

**PICT response in this soil.**

We have removed the phrase "supporting the absence of a PICT response" in line 146. Also we have introduced the concept of "cost of tolerance" after line 142, as explained before in response to the reviewer comment.

**L147 and L155: Can you please check Table numbers? I believe it's Table 3.**

As indicated by the reviewer it is Table 3 and not Table 2.

**L152: "does not necessarily result"**

Ok, the error was corrected.

**L161: Tables 3 and 4.**

Ok, the error was corrected.

**L173: Maybe "even though" instead of "even that"?**

We agree with the suggestion.

**Figure 3: I'm sorry, what do lowercase letters mean? Replicates within treatments sometimes have different lowercase letters, e.g., CD/SG_a, CD/SG_b and CD/SG_c.**

Lowercase letters were used to identify the different replicates within treatments.

Anonymous Referee #2

**The response of microbial communities to different perturbations is of great interest for designing sustainable farming practices (either tactic or strategic). Particularly the long term effect of GBHs is relevant in no-till agricultural systems, and the dry-wetting effects are important in rainfed agriculture. This research explores in a microcosm experiment the effect of GBHs and dry-rewetting perturbations on soil microbial communities, but the interaction effect was not clear. Despite sound methods were used in the present work, deeper studies are needed and**

**can be addressed with new research techniques like microbiome sequencing and also by repeated cycles of dry-rewetting to address more clearly the ecological impact (eg. resilience, resistance to disturbance). The manuscript is appropriate for publishing in SOIL. Some minor corrections are needed:**

**1- Check references: year in text is different from the year in References list a. Line 40 and 148: Evans and Wallenstein, 2011 or 2012? b. Line 87: Zabaloy et al 2016 is not in Reference list c. Line 89: Pfeiffer et al 2013 or**

**2014? d. Line 144: Clements and Rohr 2009 or 2008? e. Line 151: Franzluebbers et al 1995 or 1994?**

All references were checked and the modifications were introduced as indicated by the reviewer.

**2- As Reviewer #1 suggests, the concept of PICT response and the absence of interaction could be explained with more detail.**

We agree with the need of clarification of this concept. Please see the response to the third comment of Reviewer 1 (L145-146).

**3- Line 58: how many years is "long term"? Despite described in Allegrini et al 2015, please indicate in the text.**

With long-term we refer to a history of more than 20 years. We introduced it in the text as suggested by the reviewer.

**4- Line 48: change quantitae by quantity.**

The change was introduced in the text as indicated by the reviewer.

**List of all relevant changes**

All relevant changes were introduced in response to the reviewers' comments and were indicated previously.

In addition to these changes, the following changes were also introduced:

**Abstract (Lines 23 to 25 of the revised version of the manuscript)**

The following lines were introduced in response to the Topical Editor comments:

"The results of this study indicate that the perturbation of an acute glyphosate exposure in a soil with long-history of this herbicide does not have a conditioning effect on the response to a subsequent dry-rewetting disturbance according to a physiological indicator or the quantified bacterial/archaeal genes."

**Data availability**

A typing error was found in the name of the database and was corrected:

"4TL Database" changed by "4TU Research.Data Database"

**Response to Executive Editor**

**1. In the editor comments I requested that you revise the final statements in the abstract to better articulate the new findings presented and the wider implications of these results. You have added a sentence which nicely summarises the results but you have not described the wider implications. For a wider audience a comment on the relevance of these findings to sustainable agriculture under future climate change would broaden the appeal of the paper. A comment on this in the conclusions would also be appropriate.**

The following lines on the relevance of our findings to sustainable agriculture were added in the abstract and in the conclusion:

Abstract:

"This is particularly relevant for the sustainability of soils in rainfed agriculture, where frequent exposure to GBHs along with intensification of hydrological cycles are expected to occur. Further studies considering multiple dry–rewetting disturbances and in different soil types should be conducted to simulate those conditions and to validate our results."

Conclusion:

"In a global context of extended use of GBHs and climate models which predict an intensification of hydrological cycles, our results are particularly relevant for the sustainability of soils in rainfed agriculture, where dry-rewetting cycles and GBHs applications are expected to occur simultaneously."

**2. The second point is in the methods you have not included detail of the field site but have referenced another paper. Please include the location of the field site and the soil type here. Apologies I did not notice this on the earlier review.**

The location and the soil type were added as requested.